# Neutral Human Milk Oligosaccharides Are Associated with Multiple Fixed and Modifiable Maternal and Infant Characteristics

**DOI:** 10.3390/nu12030826

**Published:** 2020-03-20

**Authors:** Meichen Wang, Zhenwen Zhao, Ai Zhao, Jian Zhang, Wei Wu, Zhongxia Ren, Peiyu Wang, Yumei Zhang

**Affiliations:** 1Department of Nutrition and Food Hygiene, School of Public Health, Peking University, Beijing 100191, China; meichenw@outlook.com (M.W.); zhangjiantp@163.com (J.Z.); wennie0616@163.com (W.W.); renzhongxia@bjmu.edu.cn (Z.R.); wpeiyu@bjmu.edu.cn (P.W.); 2Beijing National Laboratory for Molecular Sciences, Key Laboratory of Analytical Chemistry for Living Biosystems, Institute of Chemistry Chinese Academy of Sciences, Beijing Mass Spectrum Center, Beijing 100190, China; zhenwenzhao@iccas.ac.cn; 3Department of Social Medicine and Health Education, School of Public Health, Peking University, Beijing 100191, China; aizhao@bjmu.edu.cn

**Keywords:** human milk oligosaccharides, secretor, lactation stage, influenced factors

## Abstract

We aimed to identify if maternal and infant factors were associated with neutral human milk oligosaccharides (HMOs) variability and examined the associations between HMOs concentration and infant growth and disease status in healthy Chinese mothers over a 6-month lactation period. We recruited mothers and their full-term infants as our subjects. At 1–5 days, 8–14 days, 4 weeks, and 6 months postpartum, all participants were interviewed to collect breast milk samples, obtain follow-up data and measure infant length and weight at their local hospital. A total of 23 neutral HMOs were analyzed by high performance liquid chromatography (HPLC)- mass spectrometer (MS). Secretor and Lewis phenotype were determined by the concentration of 2′-fucosyllactose (2′-FL) and Lacto-N-fucopentaose (LNFP)-II. The associations between maternal and infant factors with HMOs concentrations were investigated. A total of 464 human breast milk samples were collected from 116 mothers at four different time points. In total, 76.7% mothers were found to be Secretor and Lewis positive phenotype (Se+Le+), 17.2% were Se-Le+, 4.3% were Se+Le-, and 1.7% were Se-Le-. Several individual HMOs, including 2′-FL, Lactodifucotetraose (LDFT), LNFP-I were determined by Secretor phenotype. Most individual HMOs decreased at the later stage of lactation, except 3′-FL. We suggest that Secretor phenotype and lactation stage could influence most of the neutral HMOs. Concentrations of specific HMOs may be associated with maternal age, allergic history, pre-pregnancy body mass index (BMI), parity, delivery mode, infant gestational age and gender.

## 1. Introduction

Human milk is widely acknowledged as nature’s first functional food [1]. In addition to providing nutrition for nearly all infants, this miraculous fluid contains a myriad of bioactive components, including hormones, immunoglobulins and oligosaccharides [2]. HMOs are complex carbohydrates that humans can’t digest but Bifidobacterium and other beneficial bacteria thrive on them [3]. HMOs are the third most abundant solid component in human milk, after lactose and lipids. These substances have a variety of benefits, including stimulating growth of beneficial intestinal bacteria [3], modulating the immune system of the intestinal mucosa [4,5,6], protecting against infection [7,8] and promoting postnatal brain development [9].

HMOs are typically constituted of three to ten monosaccharide units, consisting of glucose (Glc), galactose (Gal), N-acetyl-glucosamine (GlcNAc), fucose (Fuc), and sialic acid (Neu5Ac) [10]. Recently, researchers have discovered that approximately 150 oligosaccharide structures in human milk have been identified [11]. The core group of HMOs present at the reducing end is either lactose or N-acetyl-lactosamine (LacNAc) [12]. The core group can be further decorated with fucose residues by the action of fucosyltransferases (FUTs), and sialic acid residues by the action of sialylltransferases [13]. HMOs can be classified into three main groups (neutral core, neutral fucosylated, and acidic) according to the monosaccharides present in the structure [13,14]. Generally, neutral HMOs account for more than 75% of the total HMOs [6]. 

HMOs are produced in the human mammary gland and synthesized by competing enzymes. A broad variability in the pattern of HMOs showing in different individuals is well known [11]. There are several studies reporting that Secretor-status and Lewis type can determine fucosylation patterns [15]. FUT2 encoded by the Secretor gene determines the presence of α1,2-fucosylated HMOs such as 2′-FL and LNFP-I, and FUT3 encoded by the Lewis gene synthesizes α1,4-fucosylated HMOs such as LNFP-II and Lacto-N-difucohexaose (LNDFH) [15,16,17]. Other FUTs determined the presence of α1,3-fucosylated HMOs such as 3′-FL and LNFP-IV, whose synthesis was not influenced by the Se or Le genes. Therefore, there are four major patterns of HMOs according to the activities of FUT2 and FUT3. The distribution of HMOs patterns in different populations varies depending on genetic background (Se+Le+, Se-Le+, Se+Le- and Se-Le-) [18,19]. 

Beyond genetic factors, several studies reported that HMOs composition varied with lactation stage [11,15,17,20,21,22,23,24,25,26]. However, most studies collected samples in a single time point or focused on first few weeks of lactation. In addition, HMOs vary during lactation and may also be influenced by the gestational stage [17,27]. Recently, researchers found that HMOs composition and concentration presented a wide variability in women with the same genetic background [28], which suggested that sociodemographic and environmental factors may play a role. At the same time, little is known about the effect of other non-genetic factors on the composition and concentration of HMOs. It is important to understand the influencing factors on HMOs since some of the HMOs play a crucial role in infants’ health. Specific HMOs may be associated with infant nutritional and disease status [5,29].

We recruited postpartum women within 5 days of delivery in hospitals and collected human milk samples at 1–5 days, 8–14 days, 4 weeks and 6 months postpartum, respectively. Maternal and infant characteristics were obtained from mothers. The objective of this longitudinal research was to identify maternal and infant factors associated with neutral HMOs variability and examine the associations between HMOs concentration and infant growth and disease status in healthy Chinese mothers over the first 6 months of lactation.

## 2. Materials and Methods 

### 2.1. Design and Study Population

This prospective study was embedded in the Maternal and Infants Nutrition Cohort Study which was performed in three cities (Beijing, Xuchang and Suzhou) of China between December 2017 and December 2018. We selected one Maternal and Child Health Hospital or Community Health Center in each city, where we recruited mothers and their infants at 1–5 days after delivery. The eligible criteria for mothers included, (1) must be ≥18 years of age, (2) having a self-reported health status, (3) breastfeeding a full-term infant (gestational age ≥37 weeks), (4) and being resident in one city for more than one year. At 1–5 days, 8–14 days, 4 weeks (27–33 days), and 6 months (177–183 days) postpartum, all participants in the study were interviewed to collect breast milk samples, obtain follow-up data and measure infant length and weight at their local hospital. The follow-up appointment was scheduled to coincide with their routine postnatal assessments or their child’s vaccination schedule. We provided reminder phone calls before the mothers’ scheduled appointment. If they could not come to the research site, a home visit was conducted. A trained interviewer or professional nurse followed and supported the subjects during the study period. Initially, 269 women agreed to participate in the study. Due to attrition, a total of 116 women remained in the study, and provided breast milk samples at four timepoints (Figure 1).

### 2.2. Data Collection and Variable Definitions

Basic information—including socio-demographic characteristics (age, nationality), obstetric characteristics (pre-pregnancy weight, gestational age, parity and mode of delivery) and self-reported allergy history and gender of their infant—was collected by trained interviewers or professional nurses through a validated questionnaire. Maternal height was measured using a portable stadiometer at 1 month postpartum. Infants’ weight and length were recorded at delivery. The Infant/Child ShorrBoard was used to measure infant length. Weight was measured using a portable scale with the child’s clothing, shoes, and diapers removed. We recorded the disease status of the infants, including infantile eczema, respiratory system and digestive system diseases, at 1-month and 6-month. Parity was categorized as either primiparous or multiparous. Pre-pregnancy BMI was calculated as self-reported pre-pregnant weight (kg) divided by measured height squared (m^2^). The disease status was classified as ‘Yes’ or ‘No’ according to disease information we collected.

### 2.3. Milk Collection and Preservation

Breast milk sampling was standardized for all participants. Mothers were contacted by phone and were asked to feed their infants before 7:00 a.m. on their appointment day. The milk samples were generally collected in the morning between 09:00 and 11:00 to avoid circadian influence. First, research personnel or the mother cleaned the study breast using clean water and gauze. Then, full milk expression from a single breast was collected in a collecting pipe and reversed 5–10 times for homogenization by professional nurses. Hand expression was encouraged, and pump expression was also accepted depending on cultural acceptability. Aliquots of 15 mL for colostrum (1–5 days) and transition milk (8–14 days) and 40 mL for mature milk (1 month and 6 month) were secured for our study purposes. The remaining milk was returned to the mother for feeding her infant.

The breast milk was transported to the local hospital. Each sample was distributed in 1 mL freezing tubes, labeled with the subject number and stored in a freezer. Ultimately, all samples were transported to Beijing and stored at −80 °C prior to oligosaccharide extraction.

### 2.4. HMOs Analysis Based on HPLC-ESI-MS

HMOs were extracted from human milk samples and reduced according to previous publications [30,31]. An Ultimate 3000 HPLC system (Dionex UltiMate 3000) coupled to an Orbitrap MS (Thermo Scientific Orbitrap Fusion Lumos) equipped with an electrospray ionization (ESI) source was used for the analysis. One HMO (Lacto-N-tetraose, LNT) was quantified against genuine standards with known purity. The characterization of HMOs structure and concentration can be challenging, especially because the reference standards for many HMOs are not commercially available [32]. As was performed in a previous study [17], all other HMOs were quantified against LNT in this study, assuming equimolar response factors. Although the concentration was not an absolute quantification, the representative concentration could reflect the trend and relative amounts of HMOs to some extent. To ensure the method was performing consistently between analytical batches, a pooled human milk sample (25 μL per sample) was analyzed with every batch of analysis and at least every 25 injections as a quality control (QC) sample.

#### 2.4.1. HMOs Reduction and Purification

An amount of 50 μL of milk in 150 μL of deionized water was defatted via centrifugation at 3220× *g* for 30 min. Then 50 μL of the middle liquid was transferred to an EP tube (0.5 mL), 100 μL ethanol was added for protein precipitation at −80 °C for 1.5 h. The solution was thawed, and further centrifugated at 10,000× *g* for 10 min for separation of protein. Then, 90 μL of the supernatant was transferred to a new microtube, which was vacuum frozen centrifugated until dry. Quantities of 25 μL deionized water and 25 μL 2.0 M NaBH4 were added and mixed in a 65 °C incubator for 1.5 h, this made the extracted HMOs reduce to their alditol form. 

Then, the product was desalted and purified by solid phase extraction (SPE) using nonporous graphitized carbon cartridges (GCC) [33]. Prior to use, the GCC was conditioned by deionized water, then 80% acetonitrile (ACN) in water (*v/v*) with 0.1% trifluoroacetic acid (TFA, *v/v*) and then deionized water. HMO samples were loaded onto the GCC and washed with deionized water 5 times to remove the salts and small molecule peptides. Then, the samples were eluted with 0.2 mL of 40% ACN in 0.05% TFA (*v/v*), twice. The eluents were combined in a new microtube and dried in the vacuum refrigerated centrifugal drier. Before MS analysis, 50 μL deionized water was added and the products were reconstituted to appropriate concentration.

#### 2.4.2. HMOs Identification and Quantification

The HMOs were separated on an HPLC instrument with analytical column (150 mm × 1 mm,5.0 μm particle size, Thermo Hypercarb). The mobile phases consisted of ultrapure water(A) and ACN(B), both containing formic acid (FA) at 0.1% and delivered at a flow rate of 0.2 mL/min. A 45 min gradient was used for separation [31,34]. The software Xcalibur was used to identify and quantitate free HMO by matching retention times and exact oligosaccharide masses. Twenty individual HMOs, including 2′-FL, 3′-FL, LDFT, LNFP-I, LNFP-II, LNFP-III, LNFP-IV, LNDFH-I, LNDFH-II, Lacto-N-hexaose (LNH), Lacto-N-neohexaose (LNnH), Monofucosyl-para-lacto-N-hexaose-IV (MFpLNH-IV), Isomer 1-fucosyl-paralacto-N-hexaose-I (IFLNH-I), IFLNH-III, Difucosyllacto-N-hexaose-b (DFLNH-b), Trifucosyllacto-N-hexaose-I (TFLNH-I), TFLNH-II, Difucosyl-para-lacto-N-hexaose-I (DFpLNH-I), DFpLNH-II, and Difucosyllacto-N-octaose-I (DFLNO-I), were clearly separated. However, LNT and Lacto-N-neotetraose (LNnT), Monofucosyllacto-N-hexaose-I (MFLNH-I) and MFLNH-III, DFLNH-a and DFLNH-c were not separated well. Thus, their concentrations were given as the sum of LNT+LNnT, the sum of MFLNH-I+ MFLNH-III, the sum of DFLNH-a+ DFLNH-c, respectively.

### 2.5. Ethical Considerations

This study was approved by the Medical Ethics Research Board of Peking University (No. IRB00001052-19040) and complied with the Declaration of Helsinki. Written informed consent was obtained from all participants at 1–5 days after childbirth.

### 2.6. Secretor and Lewis Phenotype Determination

Women whose HMOs contained α1,2-fucosylated structures were distributed as Secretor positive (Se+) group, others were classified as Secretor negative (Se-) group. Those who presented α1,4-fucosylated HMOs were distributed as Lewis positive (Le+) group, others were classified as Lewis negative (Le-) group. As previous studies have done [17,24,35], we chose 2′-FL and LNFP-II as representative HMOs for α1,2- and α1,4-fucosylated HMOs. 2′-FL and LNFP-II fell steeply in Se- and Le- group, respectively. The criteria are shown in Table 1.

### 2.7. Statistical Analyses

Descriptive statistics and association analysis were performed with software of SPSS version 20.0 (International Business Machines Corporation, Armonk, NY, USA) and R (version 3.4.4). Geometric means and standard deviations were used to depict continuous variables, and Numbers (proportion) were used to depict categorical variables. The characteristics description was stratified by secretor status. One-way analysis of variance (ANOVA) was employed to verify differences among Se+/Se- group. Chi-square test was conducted to compare the difference of categorical variables among Se+/Se- group. If the group number was small, Fisher’s exact test was used to verify associations. As the normality assumption was not satisfied in the HMOs concentration, significant differences between secretor status (Se+/Se-) and follow-up time points were assessed by Mann–Whitney U test and Friedman test.

Spearman rank correlations were performed to validate associations between HMOs concentrations and maternal and infant characteristics, including maternal age, allergic history, pre-pregnancy BMI, parity, delivery mode, infant gestational age and gender. Since we just had seven participants assigned to Lewis negative group in our cohort, the associations analyzed only separately for secretor status. Considering the inherent correlations of samples from same mothers, linear mixed model was used to examine the associations between neutral HMOs and maternal and infant characteristics. To correct for non-normal distributions, a log logarithmic transformation was performed to HMOs concentration. Initially, random intercept model without any covariates was run to test random intercept effects. We then studied relative factors and added an indicator variable of follow-up time points into the model. Spearman rank correlations were also performed to validate associations between HMOs concentrations and infant growth and disease status. All statistical analyses were considered statistically significant when with a *p*-value < 0.05, and all tests were performed two-tailed.

## 3. Results

### 3.1. Characteristics of Studied Population

In this study, 269 women were enrolled, and 116 were included in the analysis framework (Figure 1). A total of 464 human breast milk samples were collected from 116 mothers at 4 different time points (1st–5th day, 8st–14st day, 1st month, and 6st month postpartum). A total of 23 neutral HMOs, including 20 individual HMOs, and 3 HMO combinations, were monitored based on accurate mass and reproducible retention times. In total, 89 (76.7%) mothers were found to be Se+Le+, 20 (17.2%) were Se-Le+, 5 (4.3%) were Se+Le-, and just 2 (1.7%) were Se-Le-.

Table 2 summarized the maternal and infant characteristics according to maternal Secretor status. No significant difference among the Se+/Se- groups were observed for all the studied variables.

### 3.2. HMOs Concentration in Secretors versus Non-Secretors

Here, we found LNT&LNnT (most were LNT, accounting for approximately 50%–80%) and LNFP-I to be the most abundant neutral HMOs in milk samples. 2′-FL, LDFT, LNFP-I, LNDFH-I, IFLNH-I, TFLNH-I, TFLNH-II, DFLNH-a&c had an extremely low concentration at four follow-up time points in Non-Secretors. However, the concentration of LNT&LNnT, the core structure in human milk, was significantly higher in Non-Secretors compared Secretors. In addition, for Non-Secretors, higher amounts were found for 3′-FL, LNFP-II, LNFP-III, LNFP-IV, LNDFH-II, MFLNH-I & III, DFLNH-b, DFpLNH-I, DFpLNH-II, DFLNO-I (*p* < 0.05) in colostrum, transitional, and mature milk as compared to Secretors (Figure 2).

### 3.3. Changes in Individual HMO Concentration During Lactation

For both Secretors and Non-Secretors, the concentration of several HMOs decreased at the later stage of lactation (Figure 2). The highest concentration of LNT&LNnT, LNnH, MFpLNH-IV and DFpLNH-II was in colostrum. The highest concentration of LNFP-III, LNH, MFLNH-I & III and DFLNO-I was in transitional milk, and IFLNH-III, DFLNH-b was in 1-month mature milk. However, 3′-FL reached its maximum concentration in 6-month mature milk, and reflected an increasing trend.

For Secretors, the highest concentration of LNFP-I, IFLNH-I, and DFpLNH-I was in colostrum; LNFP-IV, LNDFH-I, LNDFH-II, TFLNH-II, and DFLNH-a&c was in transitional milk; 2′-FL was in 1-month mature milk. The concentration of LNFT was found an increasing trend.

### 3.4. Other Factors and Individual HMO Concentration

Variations in maternal age, allergic history, pre-pregnancy BMI, parity, delivery mode, infant gestational age and gender were associated with several HMOs (Table 3). For example, maternal age was negatively associated with LNT&LNnT, LNFP-I, LNFP-II, LNFP-III, LNDFH-I, DFpLNH-II and DFLNO-I, but was positively correlated with 2′-FL and DFLNH-b in Secretors. Gestational age was negatively associated with IFLNH-I and TFLNH-II in Secretors and 3′-FL, LNFP-II, LNFP-IV, DFpLNH-I and DFpLNH-II in Non-Secretors. DFLNH-b and DFpLNH-I were lower in Non-Secretors women with allergy history compared to women without allergy history. A negative correlation between parity and LDFT, LNFP-I, LNFP-IV, LNDFH-I, LNDFH-II and DFpLNH-I was found in colostrum. Compared with boys’ mothers, girls’ mothers had lower concentrations of LNFP-I, LNH, LNnH, IFLNH-I, IFLNH-III and DFLNH-a&c.

### 3.5. Associations of Individual HMO with Infant Growth Status and Disease Status

Significant associations occurred among some individual HMO and infant growth status, especially in Secretors (Table 4). For Secretors, infant length gain at month1 was positively associated with LNH, LNnH, MFpLNH-IV, IFLNH-I and DFLNH-a&c in colostrum, TFLNH-I and DFLNH-b in transitional milk, and LNnH, MFpLNH-IV, IFLNH-III, TFLNH-I, TFLNH-II and DFLNH-b in mature milk. Infant disease status was associated with IFLNH-III in colostrum, and LNnH, IFLNH-I and IFLNH-III in transitional milk.

## 4. Discussion

Here, we found that several individual HMOs were determined by the Secretor phenotype, and most of HMOs decreased at later stage of lactation, except 3′-FL and LNFT. We also focused on the fixed and modifiable factors, including maternal age, allergic history, pre-pregnancy BMI, parity, delivery mode, infant gestational age and gender. In addition, the associations between HMOs concentration and infant growth status and disease status were examined.

### 4.1. Analysis Method of HMOs

The structures, concentrations, and patterns of HMOs have been popular topics for research in recent years [36]. These components are partially responsible for the special character of human milk. The unique diversity of HMOs contains some infrequent, complicated, and high weight structures. However, the identification and elucidation of these complex structures are analytical challenges. Currently, a variety of techniques are applied to characterize oligosaccharides structures. For instance, HPLC-MS [22,37,38], capillary electrophoresis (CE) [39], LC-MS with the fluorous derivatization method [23], gas-phase spectroscopy [12], and hydrophilic interaction liquid chromatography (HILIC)-ESI-MS/MS [32] have developed to separate and quantitatively determine several oligosaccharides. Most of the quantitative methods are based on HPLC-MS. In this study, a highly effective and sensitive method, the HPLC-ESI-MS-based method, was used for profiling milk oligosaccharides. Lots of oligosaccharides were able to be screened, and a total of 23 neutral HMOs, including 20 individual HMOs and 3 HMO combinations, were monitored.

### 4.2. Secretor and Lewis Phenotype

Fucosylation patterns of HMOs were determined by Secretor status and Lewis type [15,16,17]. The frequency of the SeLe phenotypes vary ethnically [18,29,40]. In our cohort, the Secretor and Lewis phenotypes were determined by the concentration of 2′-FL and LNFP-II as previous studies have also done [17,24,35]. We found that 76.7% mothers were Se+Le+ phenotype, 17.2% were Se-Le+, 4.3% were Se+Le-, and just 2 (1.7%) were Se-Le-. The result was similar with a genes polymorphisms study in China [41], which revealed that 9.8% of individuals were the Le- type and 22.5% were the Se- type. 

### 4.3. HMOs Concentration in Secretors versus Non-Secretors

Our study showed several individual HMOs, including 2′-FL, LDFT, LNFP-I, LNDFH-I, IFLNH-I, TFLNH-I, TFLNH-II, DFLNH-a&c, were determined by Secretor phenotype, supporting previously reported results [15,21,42]. These HMOs were regard to α1,2-fucosylated HMOs, synthesized by FUT2 which encoded by Secretor gene. The absence of FUT2 explains the extremely low concentration of these α1,2-fucosylated HMOs. Despite the concentration of fucosylated HMOs was low in Non-Secretors, the concentrations of LNT&LNnT, 3′-FL, LNFP-II, LNFP-III, LNFP-IV, LNDFH-II, MFLNH-I & III, DFLNH-b, DFpLNH-I, DFpLNH-II, and DFLNO-I were higher than Secretors. Albrecht et al. [43] reported that LNT represented a higher percentage of the total oligosaccharide-content in the non-Secretors profile (33% ± 2%) compared to the Secretors (9% ± 5%). The average concentration of LNT can be used as extra information next to α1,2-fucosylated or α1, 4-fucosylated HMOs to distinguish the SeLe phenotype [15]. Although LNT and (LNnT) were not separated well in present study, LNT accounts for approximately 50%–80%. The LNT was higher in Non-Secretors than that in Secretors. Mothers with different genetic backgrounds possessed different patterns of HMOs [16,35]. Because of the absence of α1,2-fucosylated HMOs, neutral nonfucosylated HMO might play a compensational function, which may explain that there are several benefits of HMOs for all phenotype.

### 4.4. HMOs Concentration During Lactation

The human breast milk samples were collected at four time points. Some of the dynamics in individual HMO concentration during lactation could be observed. Most of the HMOs decreased at the later stage of lactation except 3′-FL. A decreasing trend in the concentration of 2′-FL, LNT, LNnT, especially from 1 to 2 months, was observed by Sprenger et al. [20], and LNFP-I concentrations decreased significantly during the first months of lactation [11]. This may reflect a unique role for these HMOs in the early stage of infant as requirements for growth and immunity [14]. For instance, some HMOs may act in the prevention of gut dysfunction and necrotizing enterocolitis [7,44]. Furthermore, a significant increase in 3′-FL concentrations during the first months of lactation was also reported in some studies [11,17,24]. In this study, we also found a significant increase in LNFT concentrations. These HMOs may play an important role in the later stage of infant development in terms of microbiota and gut maturation [45]. However, the biological relevance of these trends is still unclear, and would be worthy of future investigation. 

Our analysis revealed that some individual HMOs do not show a significant increasing or decreasing tendency during the first month. The highest concentrations of LNFP-III, LNH, MFLNH-I & III, and DFLNO-I were in transitional milk. Nevertheless, these HMOs deeply decreased at 6 months. One study revealed that fucosylated HMOs could regulate the gut microbiome of infants during different lactation stages [27]. We believe that infants need to be exposed to a certain content of HMOs at specific stages of their development, and different pathways of HMOs exert different roles during different stages. 

### 4.5. Other Factors & HMOs

In the present cohort, the studied variables were irrelevant to the Se+/Se- group, but relevant to some specific individual HMOs. 

For Secretors, maternal age was negatively associated with several HMOs, including LNT&LNnT, LNFP-I, and positively correlated with 2′-FL and DFLNH-b. McGuire et al. [18] reported that age was negatively correlated with concentrations of LNnT (r = −0.14). This result may relate to physical change caused by age. However, the extent of the influence is still uncertain.

There are several hypotheses about the role of HMOs on infant allergy [46]. However, little study has focused on the relationship between maternal allergy history and HMOs, Tonon et al. [24] observed significantly higher DFpLNnH concentrations in the milk from Secretors with allergic diseases than that of those without allergic diseases. In the present study, DFpLNH-I and DFpLNH-II were related to maternal allergy history. We assumed that maternal allergy history could explain, at least in part, the variability in HMOs. Maternal BMI was positively correlated with total HMOs and 2′-FL [15,18]. Our study found that pre-pregnancy BMI was positively correlated with LNH, LNnH, IFLNH-I and DFLNH. 

Few studies have examined the associations between parity and HMOs, Karina M et al. [24] observed a significant positive correlation between parity and LNT+LNnT, and a negative correlation between parity and 3′-FL in Se+Le- mothers. Interestingly, we only found a negative correlation between parity and LDFT, LNFP-I, LNFP-IV, LNDFH-I, LNDFH-II and DFpLNH-I in colostrum. 

Austin et al. [17] observed that significant differences existed between preterm and full-term infant milk at equivalent adjusted gestational ages. Although our subjects were full-term infants, we also found that gestational age was negatively associated with IFLNH-I and TFLNH-II in Secretors and 3′-FL, LNFP-II, LNFP-IV, DFpLNH-I, DFpLNH-II in Non-Secretors. 

Concentrations of some composition in human milk differed in relation the infant’s sex [47,48]. In secretors, mothers of boys had higher concentrations of LNFP-I, LNH, LNnH, IFLNH-I, IFLNH-III and DFLNH-a&c than those of girls. The difference in perinatal outcomes between males and females has been recognized as the “male disadvantage” since the 1970s [49]. Boys may need more HMOs than girls, and this theory needs further studies.

### 4.6. HMOs & Infant Growth and Disease Status

Studies investigating how HMOs are related to infant growth are sparse. In the present study, there was no significant difference in infant weight and length gain between the Se+/Se- groups. This result is consistent with previous studies, which reported that the relatively substantial variation in HMOs between Secretors and Non-Secretors did not impact the infant growth of either sex up to 4 months of age [20]. However, David et al. [50] reported that differences in HMO composition in human milk were associated with infant body composition, especially body fat deposition. Our study also found some individual HMOs were associated with infant growth status, especially in Secretors. For HMOs, the association with infant growth needs further investigation. HMOs concentration was also associated with infant disease status. This may relate to HMOs’ effects on the health of the development of immune system [6] and the establishment of the gut microbiota [51]. 

### 4.7. Strengths and Limitations of the Study

To the best of our knowledge, this is the first large cohort study characterizing the change and influencing factors of the HMO concentration for a population of the Chinese mainland. The design is extremely important, as the follow-ups were done at 1–5 days, 8–14 days, 4 weeks, and 6 months postpartum. The same investigator/nurse collected the questionnaires and samples from one mother to ensure the uniformity and accuracy of the information obtained. In addition, the analytical method used for determination of the HMOs has been extensively validated. However, a major limitation of present study was that we did not analyze sialylated HMOs because of destruction for pretreatment. Another limitation was that LNT and LNnT were not separated well in the present analysis. Therefore, their concentration was given as the sum.

## 5. Conclusions

In conclusion, our longitudinal study provided comprehensive data on neutral HMOs from breastfeeding Chinese women. We revealed that Secretor phenotype and lactation stage could influence most of the neutral HMOs. Concentrations of specific HMOs might associate with maternal age, allergic history, pre-pregnancy BMI, parity, delivery mode, infant gestational age and gender. Different Secretor phenotypes were irrelevant to maternal and infant factors in the present study, and did not impact infant growth.

## Figures and Tables

**Figure 1 nutrients-12-00826-f001:**
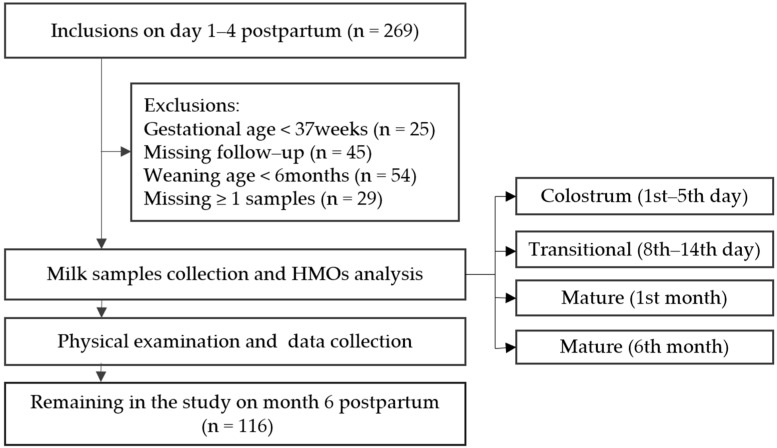
Flow chart of inclusions.

**Figure 2 nutrients-12-00826-f002:**
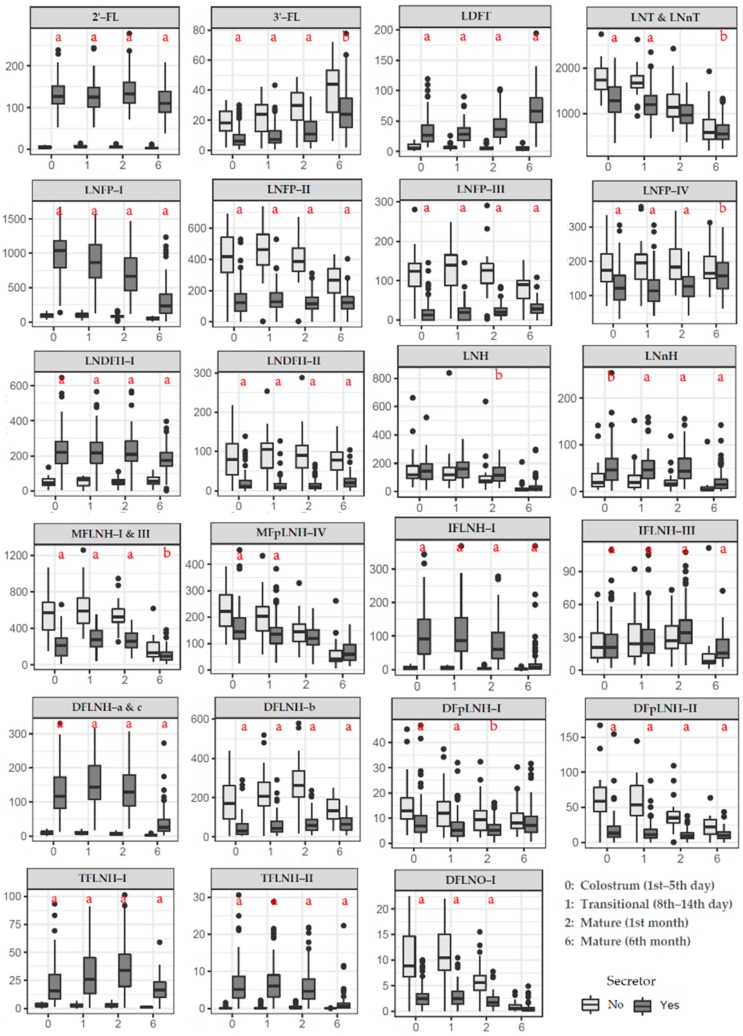
Mean concentration of HMOs during different lactation periods for secretors and non-secretors, mg/L. Letters indicate if difference between Secretors and Non-Secretors is significant; a: *p* < 0.001, b: *p* < 0.05.

**Table 1 nutrients-12-00826-t001:** Assignment of mothers to groups according to 2′-FL and LNFP-II.

	2′-FL		LNFP-II
Se+	≥15 mg/L in ≥1 milk sample	Le+	≥30 mg/L in ≥1 milk sample
Se-	<15 mg/L in all milk samples	Le-	<30 mg/L in all milk samples

**Table 2 nutrients-12-00826-t002:** Characteristics of study population, segregated by Secretor status. Mean ± SD or N (Percentage).

	All Women (n = 116)	Secretors (n = 94)	Non-Secretors (n = 22)	*p*
**Mothers**				
Age, years	29.92 ± 5.18	29.76 ± 4.47	30.63 ± 7.6	0.479
Area				0.403
Beijing	59 (50.9%)	47 (50%)	12 (54.5%)	
Xuchang	29 (25%)	22 (23.4%)	7 (31.8%)	
Suzhou	28 (24.1%)	25 (26.6%)	3 (13.6%)	
Allergic history				0.913
Yes	15 (12.9%)	12 (12.8%)	3 (13.6%)	
No	101 (87.1%)	82 (87.2%)	19 (86.4%)	
Pre-pregnancy BMI, kg/m^2^	20.96 ± 3.16	20.97 ± 3.03	20.94 ± 3.74	0.964
Parity				0.397
Primiparous	92 (79.3%)	76(80.9%)	16 (72.7%)	
Multiparous	24 (20.7%)	18(19.1%)	6 (27.3%)	
Delivery mode				0.774
Cesarean section	34 (29.3%)	27 (28.7%)	7 (31.8%)	
Natural delivery	82 (70.7%)	67 (71.3%)	15 (68.2%)	
**Infants**				
Gestational age, weeks	39.56 ± 1.05	39.51 ± 1.01	39.79 ± 1.17	0.257
Gender				0.960
Boys	68 (58.6%)	55 (58.5%)	13 (59.1%)	
Girls	48 (41.4%)	39 (41.5%)	9 (40.9%)	
Weight at birth, Kg	3.46 ± 0.42	3.45 ± 0.43	3.47 ± 0.39	0.883
Month 1 weight gain, Kg	1.31 ± 0.64	1.33 ± 0.67	1.20 ± 0.51	0.397
Month 6 weight gain, Kg	4.67 ± 1.07	4.62 ± 1.13	4.86 ± 0.83	0.366
Length at birth, cm	50.47 ± 1.65	50.37 ± 1.59	50.91 ± 1.82	0.165
Month 1 length gain, cm	4.63 ± 2.10	4.63 ± 2.13	4.60 ± 2.01	0.958
Month 6 length gain, cm	17.72 ± 2.91	17.73 ± 2.77	17.66 ± 3.46	0.923
Month 1 (sick/illness)				0.503
Yes	92 (79.3%)	74 (78.7%)	18 (81.8%)	
No	24 (20.7%)	20 (21.3%)	4 (18.2%)	
Month 6 (sick/illness)				0.391
Yes	79 (77.5%)	61 (75.3%)	18 (85.7%)	
No	23 (22.5%)	20 (24.7%)	3 (14.3%)	

**Table 3 nutrients-12-00826-t003:** Associations of Maternal and Infant Factors with individual HMO Concentration according to secretor status.

	**Maternal Age**					**Allergic History**					**Pre-Pregnancy BMI**					**Parity**				
	**0**	**1**	**2**	**6**	**P ^a^**	**0**	**1**	**2**	**6**	**P ^a^**	**0**	**1**	**2**	**6**	**P ^a^**	**0**	**1**	**2**	**6**	**P ^a^**
Non-Secretors																				
3′-FL	0.35	0.16	0.11	0.09	0.887	0.34	0.37	0.24	0.26	0.549	−0.21	−0.32	−0.51 *	−0.35	0.109	−0.11	0.05	0.08	0.03	0.766
LNT&LNnT	0.20	0.22	0.07	−0.25	0.917	0.07	−0.03	0.05	−0.03	0.969	0.30	0.17	0.26	0.18	0.363	0.19	0.11	0.11	0.14	0.393
LNFP-III	0.22	0.00	−0.01	−0.40	0.515	0.24	0.11	0.05	−0.14	0.912	−0.23	−0.08	−0.09	−0.25	0.195	0.26	0.19	0.06	0.13	0.110
LNFP-II	0.44 *	0.30	0.27	−0.20	0.327	0.43 *	0.26	0.22	0.05	0.863	−0.32	−0.24	−0.14	−0.52 *	0.125	0.11	0.06	−0.02	−0.06	0.162
LNFP-IV	0.57 **	0.29	0.33	0.08	0.284	0.22	0.03	−0.20	0.20	0.089	−0.08	−0.16	−0.18	−0.26	0.200	−0.10	−0.21	−0.23	0.19	0.925
LNDFH-II	0.34	0.25	0.13	−0.05	0.755	0.43 *	0.39	0.32	0.22	0.577	−0.19	−0.08	−0.20	−0.42	0.170	0.11	0.05	−0.02	0.03	0.694
LNH	0.00	0.15	0.16	−0.36	0.562	−0.18	−0.28	−0.09	0.03	0.147	0.08	0.00	0.13	0.03	0.176	0.08	0.05	0.08	0.10	0.731
LNnH	0.15	0.26	0.26	−0.16	0.627	−0.14	−0.20	−0.11	0.11	0.577	−0.12	−0.22	−0.11	−0.12	0.475	−0.08	−0.21	0.11	0.13	0.538
MFLNH-I & III	0.38	0.40	0.31	−0.41	0.070	−0.24	−0.30	−0.05	−0.01	0.697	0.04	0.08	0.30	0.16	0.172	−0.18	−0.10	0.08	0.02	0.688
MFpLNH-IV	0.23	0.24	0.34	−0.20	0.850	−0.05	−0.11	−0.11	0.05	0.917	0.00	−0.21	−0.01	−0.04	0.544	0.19	0.00	−0.02	0.18	0.322
IFLNH-III	0.35	0.32	0.34	−0.30	0.803	−0.18	−0.18	−0.09	0.11	0.985	−0.12	−0.25	−0.25	−0.15	0.932	−0.06	−0.18	−0.06	0.11	0.798
DFLNH-b	0.36	0.39	0.15	−0.09	0.670	0.01	0.09	0.26	0.26	0.044	−0.38	−0.35	−0.30	−0.36	0.005	−0.05	0.11	0.05	0.10	0.755
DFpLNH-I	0.42	0.08	0.31	0.09	0.444	0.43 *	0.24	−0.05	0.34	0.040	0.01	−0.27	−0.21	−0.51 *	0.136	0.06	−0.08	−0.18	0.10	0.721
DFpLNH-II	0.27	0.12	0.18	−0.20	0.485	0.45 *	0.32	0.16	0.01	0.504	−0.38	−0.35	−0.18	−0.56 **	0.038	0.21	0.04	0.01	0.13	0.286
DFLNO-I	0.24	0.19	0.14	−0.33	0.374	0.16	0.20	0.24	−0.04	0.339	−0.43 *	−0.35	−0.21	−0.42	0.197	0.27	0.32	0.20	0.07	0.504
Secretors																				
2′-FL	0.22 *	0.10	0.23 *	0.14	0.008	0.03	−0.15	0.03	0.01	0.518	0.00	0.14	0.21 *	0.00	0.186	−0.15	−0.11	−0.13	−0.02	0.178
3′-FL	−0.04	−0.10	−0.09	−0.04	0.211	−0.03	−0.06	−0.02	−0.06	0.507	−0.04	0.03	0.04	0.13	0.461	−0.08	−0.14	−0.07	−0.01	0.638
LDFT	0.12	0.07	0.14	0.05	0.830	0.20 *	−0.08	−0.05	−0.06	0.702	−0.12	0.05	0.11	0.02	0.602	−0.27 **	−0.14	−0.13	0.01	0.172
LNT&LNnT	−0.29 **	−0.42 **	−0.45 **	−0.36 **	<0.001	−0.05	0.04	0.03	0.12	0.433	−0.08	0.01	−0.05	−0.01	0.995	−0.15	0.01	0.10	0.11	0.978
LNFP-III	0.19	0.05	−0.12	−0.30 **	<0.001	−0.04	0.05	−0.02	−0.06	0.654	−0.03	−0.17	−0.18	−0.03	0.788	−0.10	−0.10	−0.05	0.09	0.216
LNFP-II	−0.10	−0.20	−0.12	−0.20	<0.001	−0.03	−0.01	−0.02	−0.02	0.509	−0.15	−0.14	−0.10	0.03	0.768	−0.14	−0.05	0.00	0.07	0.906
LNFP-I	−0.21 *	−0.29 **	−0.30 **	−0.21 *	0.043	−0.08	−0.03	0.05	0.16	0.683	−0.03	0.05	0.03	−0.04	0.941	−0.20 *	−0.03	0.04	0.13	0.290
LNFP-IV	0.07	−0.13	0.01	−0.08	0.679	0.01	−0.06	−0.07	−0.18	0.225	−0.02	0.00	0.06	0.11	0.608	−0.26 **	−0.19	−0.13	−0.13	0.027
LNDFH-I	−0.03	−0.26 *	−0.14	−0.21 *	0.021	0.05	−0.12	−0.10	−0.01	0.281	−0.13	0.04	0.06	−0.01	0.546	−0.24 *	−0.13	−0.12	0.04	0.718
LNDFH-II	−0.02	0.03	−0.01	−0.04	0.338	0.07	−0.04	−0.07	−0.07	0.887	−0.10	−0.11	−0.12	0.04	0.757	−0.23 *	−0.12	0.00	0.08	0.621
LNH	0.09	0.07	0.10	−0.08	0.680	0.04	0.06	0.11	0.20	0.269	0.13	0.24 *	0.27 **	0.08	0.043	0.18	0.14	0.20	0.10	0.131
LNnH	0.16	0.08	0.23 *	−0.07	0.141	0.04	0.03	0.03	0.11	0.607	0.18	0.27 **	0.38 **	0.11	0.004	0.15	0.10	0.19	0.05	0.295
MFLNH-I & III	0.16	0.10	0.10	−0.08	0.632	−0.11	−0.07	−0.03	0.06	0.961	0.05	0.03	0.03	0.04	0.905	0.06	−0.02	−0.02	−0.04	0.652
MFpLNH-IV	0.06	−0.08	0.10	−0.17	0.779	0.04	0.01	−0.04	0.07	0.943	0.10	0.16	0.22 *	0.10	0.047	−0.10	−0.06	−0.02	−0.01	0.552
IFLNH-III	0.24 *	0.18	0.26 **	−0.12	0.149	−0.02	−0.06	−0.02	0.07	0.957	0.13	0.13	0.24 *	0.12	0.072	0.06	−0.04	−0.02	−0.04	0.540
IFLNH-I	0.23 *	0.19	0.23 *	0.18	0.056	−0.08	−0.06	0.01	0.14	0.387	0.26 *	0.38 **	0.39 **	0.18	0.044	0.05	0.03	0.06	0.03	0.847
TFLNH-I	0.20	0.08	0.14	−0.10	0.621	−0.11	−0.15	−0.08	0.08	0.840	0.16	0.18	0.19	0.12	0.687	−0.01	−0.08	−0.15	−0.03	0.917
TFLNH-II	0.23 *	0.12	0.15	0.06	0.237	−0.10	−0.19	−0.07	0.08	0.891	0.23 *	0.30 **	0.31 **	0.08	0.746	−0.04	−0.05	−0.02	−0.01	0.666
DFLNH-a&c	0.28 **	0.16	0.19	0.01	0.161	−0.14	−0.18	−0.04	0.10	0.514	0.19	0.30 **	0.31 **	0.10	0.018	−0.05	−0.09	−0.08	−0.03	0.568
DFLNH-b	0.33 **	0.31 **	0.34 **	0.17	0.041	−0.16	−0.09	−0.09	−0.08	0.300	0.14	0.07	0.11	0.15	0.500	−0.10	−0.16	−0.17	−0.09	0.155
DFpLNH-I	0.01	−0.12	−0.04	−0.07	0.642	0.05	−0.06	−0.12	−0.16	0.198	0.00	0.06	0.11	0.11	0.294	−0.26 *	−0.18	−0.16	−0.11	0.031
DFpLNH-II	−0.16	−0.25 *	−0.26 *	−0.24 *	0.01	−0.02	−0.03	−0.03	−0.07	0.940	−0.07	−0.03	−0.07	0.04	0.876	−0.19	−0.08	−0.04	0.04	0.549
DFLNO-I	−0.05	−0.17	−0.20	−0.23 *	0.003	−0.12	−0.10	−0.08	0.07	0.666	0.03	0.03	0.07	0.05	0.752	−0.11	−0.15	−0.04	0.13	0.654
	**Delivery Mode**						**Gestational Age**							**Infant Gender**						
	**0**		**1**	**2**	**6**	**P ^a^**	**0**		**1**		**2**	**6**	**P ^a^**	**0**		**1**		**2**	**6**	**P ^a^**
Non-Secretors																				
3′-FL	0.24		0.28	0.08	0.25	0.098	−0.53 *		−0.46 *		−0.17	−0.17	0.025	0.23		0.15		0.04	0.02	0.326
LNT&LNnT	−0.21		0.07	−0.01	−0.08	0.486	−0.04		−0.25		−0.20	0.27	0.358	−0.04		0.05		0.04	−0.24	0.724
LNFP-III	0.18		0.22	0.18	0.05	0.992	−0.50 *		−0.42		−0.46 *	−0.01	0.053	0.18		0.21		0.08	−0.23	0.957
LNFP-II	0.25		0.35	0.24	0.39	0.964	−0.46 *		−0.43 *		−0.58 **	0.00	0.033	0.42		0.36		0.42	−0.02	0.100
LNFP-IV	0.12		0.24	−0.05	−0.10	0.449	−0.68 **		−.64 **		−0.61 **	0.08	0.034	0.27		0.18		0.09	−0.31	0.944
LNDFH-II	0.24		0.41	0.18	0.27	0.070	−0.32		−0.36		−0.27	0.09	0.804	0.31		0.49 *		0.42	0.02	0.535
LNH	−0.52 *		−0.41	−0.27	−0.08	0.020	0.02		−0.10		−0.06	0.41	0.296	−0.18		−0.23		0.07	−0.11	0.164
LNnH	−0.44 *		−0.32	−0.55 **	−0.15	0.016	−0.11		−0.21		−0.14	0.28	0.586	−0.28		−0.24		−0.18	−0.15	0.152
MFLNH-I & III	−0.19		−0.25	−0.30	−0.01	0.315	−0.32		−0.19		−0.13	0.34	0.700	0.21		0.09		0.24	0.02	0.460
MFpLNH-IV	−0.41		−0.10	−0.33	−0.16	0.036	−0.25		−0.43 *		−0.35	0.25	0.461	−0.27		−0.11		0.02	−0.33	0.060
IFLNH− III	−0.39		−0.35	−0.44 *	−0.19	0.023	−0.28		−0.23		−0.23	0.28	0.904	−0.21		−0.24		−0.08	−0.17	0.472
DFLNH-b	0.15		0.12	0.07	0.12	0.258	−0.49 *		−0.41		−0.24	0.09	0.457	0.31		0.28		0.28	0.05	0.611
DFpLNH-I	0.05		0.28	0.01	−0.02	0.543	−0.47 *		−0.53 *		−0.61 **	−0.01	0.010	0.05		0.07		0.09	−0.24	0.719
DFpLNH-II	0.15		0.38	0.21	0.27	0.022	−0.26		−0.43 *		−0.59 **	−0.06	0.026	−0.01		0.12		0.20	−0.17	0.191
DFLNO-I	0.18		0.20	0.30	0.29	0.396	−0.28		−0.28		−0.39	0.10	0.087	0.15		0.12		0.27	−0.06	0.337
Secretors																				
2′-FL	−0.19		−0.04	−0.05	−0.24 *	0.084	−0.08		0.14		0.01	−0.13	0.887	−0.13		−0.13		−0.21 *	−0.02	0.070
3′-FL	−0.06		0.00	−0.05	−0.04	0.878	0.04		0.03		0.04	0.05	0.264	0.07		0.02		−0.02	−0.04	0.779
LDFT	−0.20		−0.08	−0.13	0.12	0.085	0.04		0.03		0.12	0.00	0.318	0.15		−0.05		−0.04	0.09	0.569
LNT&LNnT	0.16		0.21 *	0.20	0.29 **	0.029	0.07		0.10		−0.09	−0.01	0.738	−0.16		−0.08		−0.09	−0.14	0.054
LNFP-III	−0.05		−0.03	−0.02	0.19	0.791	0.12		0.03		0.01	0.09	0.596	0.13		0.08		0.06	−0.01	0.729
LNFP-II	0.02		0.05	0.08	−0.10	0.495	0.10		0.08		0.10	0.12	0.912	0.03		0.06		0.03	−0.01	0.709
LNFP-I	0.12		0.11	0.15	−0.21 *	0.131	0.02		0.04		−0.13	−0.05	0.436	−0.15		−0.11		−0.12	−0.05	0.044
LNFP-IV	−0.05		0.06	0.08	0.07	0.625	−0.04		0.03		0.04	−0.06	0.963	−0.03		−0.08		−0.20	0.02	0.287
LNDFH-I	−0.05		0.05	0.05	−0.03	0.041	0.07		0.14		0.02	0.05	0.235	0.03		−0.02		0.02	0.11	0.138
LNDFH-II	0.09		0.08	0.00	−0.02	0.980	0.03		0.02		0.06	0.08	0.314	0.13		0.05		0.10	0.00	0.094
LNH	0.21 *		0.20	0.24 *	0.09	0.013	−0.12		−0.09		−0.15	−0.16	0.097	−0.27 **		−0.21 *		−0.21 *	−0.21 *	0.004
LNnH	0.06		0.12	0.13	0.16	0.110	−0.15		−0.06		−0.09	−0.17	0.106	−0.22 *		−0.18		−0.22 *	−0.16	0.023
MFLNH-I & III	0.12		0.11	0.17	0.16	0.133	0.03		−0.02		−0.06	−0.09	0.285	−0.13		−0.06		−0.12	−0.15	0.284
MFpLNH-IV	0.09		0.15	0.18	0.06	0.039	−0.14		−0.06		−0.11	−0.14	0.086	−0.15		−0.11		−0.19	−0.12	0.063
IFLNH-III	−0.01		0.09	0.12	0.24 *	0.233	−0.10		−0.06		0.00	−0.20	0.218	−0.18		−0.17		−0.26 *	−0.18	0.028
IFLNH-I	0.17		0.17	0.22 *	0.23 *	0.274	−0.21 *		−0.12		−0.24 *	−0.25 *	0.008	−0.34 **		−0.27 **		−0.26 *	−0.19	< 0.001
TFLNH-I	0.02		0.06	0.09	0.20	0.484	−0.06		−0.01		0.05	−0.02	0.301	−0.12		−0.08		−0.04	−0.07	0.326
TFLNH-II	0.18		0.23 *	0.25 *	0.23 *	0.451	−0.20		−0.07		−0.20 *	−0.19	0.009	−0.25 *		−0.19		−0.20 *	−0.13	0.206
DFLNH-a&c	0.18		0.22 *	0.31 **	0.28 **	0.009	−0.09		0.00		−0.08	−0.13	0.072	−0.21 *		−0.15		−0.18	−0.11	0.023
DFLNH-b	0.04		0.08	0.10	0.33 **	0.483	0.00		−0.04		0.04	−0.02	0.889	−0.05		0.01		−0.10	−0.09	0.497
DFpLNH-I	0.07		0.18	0.15	0.01	0.081	−0.11		−0.03		−0.06	−0.09	0.410	0.06		−0.09		−0.14	0.02	0.507
DFpLNH-II	0.13		0.12	0.11	−0.16	0.713	0.00		0.04		−0.03	0.02	0.841	0.09		0.04		0.02	0.04	0.383
DFLNO-I	0.25 *		0.23 *	0.28 **	−0.05	0.013	−0.04		0.04		−0.07	−0.15	0.263	−0.08		0.06		−0.08	−0.14	0.304

^a^: Linear mixed model analysis to evaluate the associations between HMOs concentration(log-transformed) and maternal age, pre-pregnancy BMI, gestational age, allergic history (ref. was no, versus yes), parity (ref. was primiparous, versus multiparous), delivery mode (ref. was cesarean section, versus natural delivery), infant gender (ref. was boys, versus girls); * *p* < 0.05, ** *p* < 0.001; 0, Colostrum (1st–5th day); 1, Transitional milk (8th–14th day); 2, Mature milk (1st month); 6, Mature milk (6th month).

**Table 4 nutrients-12-00826-t004:** Spearman rank correlation coefficient (r) between HMOs concentration and infant growth status and disease status.

	Month1 WG			Month6 WG	Month1 LG			Month6 LG	Month1 DS			Month6 DS
	0	1	2	6	0	1	2	6	0	1	2	6
Non-Secretors												
3′-FL	−0.34	−0.08	0.08	−0.44 *	−0.19	−0.13	−0.11	−0.39	−0.39	−0.26	−0.28	0.04
LNT&LNnT	0.24	0.02	0.02	0.33	0.47 *	0.47 *	0.34	0.37	−0.07	−0.06	0	0.20
LNFP-III	0.06	0.07	0.09	−0.27	0.15	0.01	0	−0.01	−0.04	0.09	−0.20	0.02
LNFP-II	−0.32	−0.27	−0.28	−0.43	−0.11	−0.16	−0.19	−0.03	−0.32	−0.19	−0.45 *	−0.22
LNFP-IV	−0.42	−0.28	−0.31	0.15	0.09	0.06	0.04	0.19	−0.32	−0.07	−0.11	0.07
LNDFH-II	−0.31	−0.39	−0.25	−0.57 **	−0.25	−0.20	−0.30	−0.10	−0.35	−0.19	−0.37	−0.22
LNH	0.20	0.05	−0.11	0.26	0.33	0.37	0.38	0.39	−0.09	−0.15	0.11	0.04
LNnH	0.06	−0.02	0.01	0.28	0.27	0.20	0.46 *	0.48 *	−0.26	−0.28	0.00	0.02
MFLNH-I & III	−0.28	−0.24	−0.20	0.25	0.07	−0.01	0.26	0.27	−0.07	0.07	0.02	0.02
MFpLNH-IV	0.19	0.01	−0.20	0.32	0.50 *	0.40	0.36	0.40	−0.32	−0.26	−0.15	0.11
IFLNH-III	−0.06	−0.07	−0.08	0.26	0.24	0.23	0.35	0.41	−0.28	−0.15	0	0.07
DFLNH-b	−0.37	−0.36	−0.30	−0.23	−0.11	−0.20	−0.19	−0.04	−0.15	−0.06	−0.15	−0.22
DFpLNH-I	−0.20	−0.04	−0.21	0.03	0.18	0.26	0.17	0.24	−0.35	−0.04	−0.19	−0.16
DFpLNH-II	−0.02	0.07	−0.14	−0.43	0.09	0.23	0.09	−0.06	−0.20	−0.11	−0.33	−0.09
DFLNO-I	0.03	0.13	0.02	−0.26	0.09	0.12	0.06	0.04	0.17	0.10	−0.02	−0.17
Secretors												
2′-FL	0.02	−0.05	−0.13	0.08	−0.08	−0.10	−0.08	0.10	−0.20	−0.12	−0.17	0.09
3′-FL	−0.05	0.06	0.09	0.27 *	−0.02	0.07	0.04	0.19	−0.06	−0.04	−0.17	−0.07
LDFT	−0.06	−0.01	0	0.28 *	−0.07	0.03	0.14	0.12	−0.06	−0.03	−0.17	−0.02
LNT&LNnT	0.13	0.04	0.07	−0.24 *	0.10	−0.05	−0.10	−0.14	0.15	0.12	0.17	0.00
LNFP-III	−0.21 *	−0.12	−0.02	−0.08	0.17	0.14	0.05	−0.13	0.10	0.08	−0.06	−0.09
LNFP-II	−0.07	−0.09	−0.04	−0.04	0.10	0.02	0.06	0.01	0.08	0.08	−0.06	−0.17
LNFP-I	0.09	0.03	0.01	−0.28 *	−0.16	−0.19	−0.23 *	−0.18	0.14	0.16	0.10	0.17
LNFP-IV	0.01	−0.02	−0.02	0.29 **	0.08	0.07	0.02	0.14	−0.11	−0.06	−0.13	−0.15
LNDFH-I	0.00	−0.03	−0.03	0.06	0.10	0	0.07	0.08	0.08	0.11	−0.04	−0.04
LNDFH-II	−0.19	−0.16	−0.03	0.04	−0.09	0.04	0.06	0.04	0.03	−0.01	−0.08	−0.15
LNH	0.02	−0.06	−0.12	−0.25 *	0.24 *	0.14	0.16	−0.06	−0.11	−0.12	0.02	−0.05
LNnH	0.09	−0.02	−0.09	−0.14	0.23 *	0.10	0.24 *	0.02	−0.16	−0.24 *	−0.07	−0.07
MFLNH-I & III	−0.04	−0.02	−0.04	−0.09	0.09	0.15	0.08	−0.09	−0.05	−0.01	0.07	−0.01
MFpLNH-IV	0.03	−0.01	−0.06	0	0.30 **	0.16	0.26 *	0.02	−0.06	−0.09	−0.05	−0.11
IFLNH-III	0.01	−0.02	−0.06	0	0.10	0.13	0.22 *	0.02	−0.22 *	−0.27 **	−0.12	−0.06
IFLNH-I	−0.02	−0.12	−0.19	−0.04	0.24 *	0.06	0.13	−0.02	−0.20	−0.22 *	−0.15	−0.01
TFLNH-I	−0.02	0.01	−0.04	0.13	0.18	0.22 *	0.24 *	0.11	−0.15	−0.16	−0.08	−0.08
TFLNH-II	−0.01	−0.04	−0.14	−0.06	0.15	0.03	0.04	−0.03	−0.17	−0.12	−0.08	0.11
DFLNH-a&c	−0.04	−0.08	−0.20	−0.08	0.21 *	0.14	0.12	−0.01	−0.16	−0.15	−0.10	0.06
DFLNH-b	−0.18	−0.15	−0.13	0.26 *	0.14	0.25 *	0.24 *	0.16	−0.14	−0.14	−0.16	−0.25 *
DFpLNH-I	−0.03	−0.04	0.01	0.27 *	0.15	0.07	0.15	0.11	−0.05	−0.09	−0.14	−0.17
DFpLNH-II	−0.05	−0.09	0.03	−0.01	0.18	0.07	0.14	−0.03	0.12	0.07	−0.02	−0.16
DFLNO-I	−0.04	−0.07	0	−0.10	0.16	0.12	0.18	−0.12	0.14	0.08	0.07	0.04

* *p* < 0.05, ** *p* < 0.001; WG, Weight gain; LG, Length gain; DS, Disease status (0 = did not get sick, 1 = get sick ≥ 1 time); 0, Colostrum (1st–5th day); 1, Transitional milk (8th–14th day); 2, Mature milk (1st month); 6, Mature milk (6th month).

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
