# Peer review of "Neutral Human Milk Oligosaccharides Are Associated with Multiple Fixed and Modifiable Maternal and Infant Characteristics"

_nutrients, 2020, doi:10.3390/nu12030826_

Round 1

Reviewer 1 Report

Please see the edits attached.

Author Response

Dear Reviewer,

Thank you for your comments concerning our manuscript entitled “Neutral Human Milk Oligosaccharides are Associated with Multiple Fixed and Modifiable Maternal and Infant Characteristics” (nutrients-741460). Those comments are valuable and very helpful for revising and improving our papers, especially in improving the accuracy and rigor of the words, avoiding produce ambiguity and making it more rigorous.

We have studied comments carefully and have made correction and increased details which we hope meet with approval.

More details shown in relative section of our manuscript. Once again, thank you very much for your comments and suggestions. Looking forward to hearing from you.

Yours sincerely,

Meichen Wang

Reviewer 2 Report

Wang et al present data examining the association between HMOs and maternal/infant characteristics. This is a really interesting topic and the manuscript is in generally nicely presented and easy to read. There are some minor issues listed below.

Abstract –

abbreviations should be defined in the first instance in the text.

Maybe 8-14 days, 4 weeks and 6 months rather than st

Introduction –

The authors provided a really nice background to the topic in simple understandable terms. They also outlined their research question really well.

There were some minor grammatical errors which should be addressed.

Methods:

The way the days are written is incorrect. Please change to 1-5 days, 8-14 days, 27-33 days and 6 months postpartum.

Some of the language is a little colloquial, this should be changed. For example line 89-90 ‘we would give them a phone call for reminder’. Just state that the visit was scheduled to coincide with routine postnatal assessments

Were details around pregnancy complications collected? I presume only mother who had a healthy pregnancy were assessed but this is not stated.

Was antibiotic use in either mother or child recorded

Were age and nationality the only socio demongraphic characteristics collected. Factors such as socioeconomic status, education level and depression scores are all known to influence the composition of breastmilk.

The authors have not stated whether there was introduction of complementary foods or supplementation with formula, this can impact milk composition and would be particularly important for teasing out effects on weight/height particularly in the 6 month samples.

7-11am – this seems like a long period of time for an infant not to feed, particularly in some of the earlier timepoints. Can the authors please discuss and comment on whether this was complied with.

Can the authors comment on the lack of standardization of collection (hand vs pump) and whether this may have had an impact.

If the authors were collecting 15ml at day 1-5 it probably wasn’t colostrum and was more likely to be transitional milk (could you comment on the ethics of taking such a high volume of colostrum during this time).

Discussion:

Overall the discussion is well thought out and reflective of the results that the authors observed. However there are some sections that are a little superficial such as the other factors and HMO section and particularly the infact outcomes section.

Author Response

Dear Reviewer,

Thank you for your comments concerning our manuscript entitled “Neutral Human Milk Oligosaccharides are Associated with Multiple Fixed and Modifiable Maternal and Infant Characteristics” (nutrients-741460). Those comments are valuable and very helpful for revising and improving our papers, especially in improving the accuracy and rigor of the words, avoiding produce ambiguity and making it more rigorous.

We have studied comments carefully and have made correction which we hope meet with approval. The main corrections in the paper and the responds to the comments are as following:

Point 1: The grammar is poor and the language is a little colloquial.

Response 1: Thank you for your encouragement. It is true that our original manuscript was not good. After proofreading, we revised the manuscript again to avoid ambiguity. All the revisions have been highlighted using the “Track Changes” function in the revised manuscript.

Point 2: Abbreviations should be defined in the first instance in the text.

Response 2: We revised the manuscript again to avoid the wrong position of abbreviations. In addition, we summarized the abbreviation at the bottom of the main section.

Point 3: Methods details about pregnancy complications, antibiotic use, depression scores.

Response 3: To be eligible for participation, women had to be have self-reported having a healthy and breastfeeding health full-term infant. In the present study, we did not consider the influence of pregnancy complications and antibiotic use. In addition, the depression scores were not collected in our study.

Point 4: Methods details about the collection of human milk samples.

Response 4: The methods of human milk samples collection specifically referenced MING study[1] which is a breast milk study in China. In practice, needs of the infants is prior to milk collection. Hand expression was encouraged. However, some mother can not accept the intimate contact, the pump expression was also accepted depending on cultural acceptability.

Point 5: Some sections that are a little superficial such as the other factors and HMO section and particularly the infact outcomes section.

Response 5: We added discussion of these section, and the changed parts are shown in our main document. More details shown in relative section of our manuscript.

The above are the point-by point corrections. Once again, thank you very much for your comments and suggestions. Looking forward to hearing from you.

Yours sincerely,

Meichen Wang

References:

  1. Austin, S.; De Castro, C.A.; Benet, T.; Hou, Y.; Sun, H.; Thakkar, S.K.; Vinyes-Pares, G.; Zhang, Y.; Wang, P. Temporal Change of the Content of 10 Oligosaccharides in the Milk of Chinese Urban Mothers. Nutrients 2016, 8, doi:10.3390/nu8060346.
